# OpenReview forum: "Towards Safe and Honest AI Agents with Neural Self-Other Overlap"
_NeurIPS.cc/2024/Workshop/SafeGenAi — SafeGenAi Oral_

### Official Review · Reviewer_UkoY · 2024-10-08
**Simple but highly effective! Excited to see such a successful empathy-based approach**

**Rating:** 10
**Confidence:** 3

**Review:**

The motivation, explanation, and implementation of the novel idea feel flawless. The approach to empathy through self-other overlap is clearly genius because it seems so obvious in retrospect. It very truthfully draws inspiration from how empathy happens in the brain. The simpleness of the approach for LLMs, the fact that it is so effective in certain contexts, and the fact that it doesn't reduce task performance somehow makes it a very strong candidate for being used widely. The fact that this paper has both language and RL results is very compelling. In particular the fact that there isn't performance decrease, as this would be difficult to justify without concrete results. I have no real feedback other than to see how it scales across diverse tasks and larger models. This paper focuses on reducing deceptions, although I believe empathy would have much broader impacts. I would be curious to know how broadly useful this approach could be for AI Alignment, and even enhancing models via social learning. If there are limitations in when it is useful for reducing deception, identifying which kinds would be very helpful. Otherwise, trying to address why it doesn't perform well in some tasks. The paper and examples were very straightforward, intuitive, and easy to grasp, despite their significance.

---

### Official Review · Reviewer_wyMK · 2024-10-09
**Applying Self Other Overlap Principles to LLM and RL agents to increase empathy and subsequently reduce self interested behaviour**

**Rating:** 5
**Confidence:** 3

**Review:**

This is a really interesting research project and a unique idea to use self-referencing from the field of neuroscience to encourage empathy.  I encourage you to do some deeper analysis and experimentation to validate the results and provide a more theoretical basis to the direction you are taking. It’s awesome that it seems to work but you need to validate it across more models and family sizes as well as draw a stronger connection between ‘empathy’ induces by SOO and deception. A formal definition of deception is needed.

**Specific Feedback**:
* The paper lacks a formal definition of deception, which is critical for contextualizing the results and the scope and intent of the work.

* Another key area for improvement is understanding the thresholds and bounds of self-other overlap (SOO). Specifically, it would be interesting to investigate whether there is a threshold of SOO where deceptive behavior consistently begins to decrease (across models and environments), as well as whether there is a recommended lower bound for SOO that ensures reduced deception.

* The reliance on a single model, Mistral 7B, which is a small language model, dramatically limits the generalizability of the findings. Testing across a range of model families and sizes is necessary to provide a stronger validation of the approach and reinforce the results. Similarly, the scope of the experiments is limited to only one reinforcement learning environment. Expanding the range of environments in which SOO is tested would help assess the robustness of the approach and subsequently strengthen the paper’s contributions.

* I'd like to see a clearer explanation of the 'burglar room' setup and why it is viewed as a form of deception. Is this classification solely based on the setup's implications for lying? Clarifying this aspect would help to connect the experiment with real-world scenarios and other work on LLM deceptive behaviors (related to the first point on the need to define what you mean by deception).

* The relationship between empathy and deception also seems ambiguous. While SOO is intended to foster empathy and thereby reduce deception,  is it possible that increased empathy could lead to more subtle forms of deception, as the model might develop a better understanding of the other agent's perspective? The connection between SOO and deception currently relies on the reduction of model complexity, but you need to substantiate this link to add some sort of theoretical depth to your approach/experimental results.

* Could the paper provide more insight into why fine-tuning was chosen over other potential steering approaches, especially considering the resource-intensive nature of fine-tuning? It may be beneficial to explore whether adjustments could be implemented at runtime through steering. Additionally, while the paper suggests this approach is more efficient than contrastive sampling techniques, could a direct comparison between fine-tuning and contrastive sampling be included to support the argument for this approach’s efficiency?

* Furthermore, I'd like to understand the approach you used to classify deceptive behavior?

* Could the paper explore potential unintended behaviors that may arise from fine-tuning agents using SOO? There are several adaptive responses or attack vectors that I can think of that could undermine the intended outcomes of SOO in promoting honest and cooperative behavior. For instance, could agents fine-tuned with SOO inadvertently use this deeper modeling to become more effective at strategic deception, especially by anticipating other agents’ reactions? In cooperative-competitive or adversarial settings, could agents exploit their SOO-based empathy by adopting behaviors that only superficially align with SOO metrics, thereby masking deceptive intents? Similarly, is there any indication that agents could manipulate their observation radius or other behavior thresholds to evade detection by staying within the "safe" SOO range? In larger multi-agent systems, is there a risk of collusion, where agents could coordinate to exploit SOO-driven empathy mechanisms for joint deceptive aims? Investigating these scenarios (or mentioning for future work) would offer a robust analysis of SOO’s potential limitations and increase our awareness on the effectiveness of this approach!

* Finally, for reproducibility, the paper should include more specific experimental details, such as whether Mistral 7B was instruction-tuned and for the cases where you did use activation data, what activation layers did you extract and why? This allows others to reproduce it but also lays out the methodological approach you took throughout your research.

* As a minor point, the definition of Representation Engineering could be updated as it does more than  modify or adjust internal processing structures; it can also be used to detect a range of behaviors such as prompt injection[1], task drift[1], and backdoors[2].
[1] Abdelnabi, Sahar, et al. "Are you still on track!? Catching LLM Task Drift with Activations." arXiv preprint arXiv:2406.00799 (2024).
[2] https://www.anthropic.com/research/probes-catch-sleeper-agents

---

### Official Review · Reviewer_arJ3 · 2024-10-10
**Novel and interesting technique for alignment**

**Rating:** 9
**Confidence:** 3

**Review:**

The paper presents a novel method, Self-Other Overlap (SOO) fine-tuning, to reduce deceptive behavior in AI systems, a critical challenge in AI safety.

Pros:
- This paper's application of neural overlap as a fine-tuning mechanism for honesty stands out as a novel and creative representation engineering technique.
- The experiments conducted on both language models (LLMs) and reinforcement learning (RL) agents show significant reductions in deceptive behavior.
- The reinforcement learning experiment was particularly interesting and creative, and provided stronger evidence for broader applicability of SOO.

Cons:
- The scope of the experiments is limited. The LLM experiments were conducted on a relatively small model (Mistral 7B) and focused on a narrow set of scenarios, such as variations of the "Bob Burglar" task. Similarly, the RL experiments were constrained to a single, simplified environment. This limitation was acknowledged in the paper.